# Insights into Epigenetic Changes Related to Genetic Variants and Cells-of-Origin of Pancreatic Neuroendocrine Tumors: An Algorithm for Practical Workup

**DOI:** 10.3390/cancers14184444

**Published:** 2022-09-13

**Authors:** Oana A. Ciobanu, Sorina C. Martin, Vlad Herlea, Simona Fica

**Affiliations:** 1Department of Endocrinology, Diabetes Mellitus, Nutrition and Metabolic Disorders, Carol Davila University of Medicine and Pharmacy, 020021 Bucharest, Romania; 2Department of Endocrinology, Diabetes Mellitus, Nutrition and Metabolic Disorders, Elias Emergency University Hospital, 011461 Bucharest, Romania; 3Department of Pathology, Fundeni Clinical Institute, 022328 Bucharest, Romania

**Keywords:** pancreatic neuroendocrine tumors, genetics, epigenetics, origins, ATRX/DAXX/MEN1, ARX/PDX1, ALT

## Abstract

**Simple Summary:**

Pancreatic neuroendocrine tumors are composite entities due to their heterogeneity illustrated in clinical behavior, mutational pattern, and site of origin. Pancreatic neuroendocrine tumors display a low mutation burden with frequently epigenetic alterations, such as histone modifications, chromatin remodeling, or DNA methylation status. Using the epigenomic data of the pancreatic neuroendocrine tumors converged to the identification of molecularly distinct subgroups. Furthermore, epigenetic signatures could be used as biomarkers due to their link to cell lineages and genetic driver mutations. We integrated the current knowledge on genetic and epigenetic alterations involved in endocrine lineage associated with these neoplasms to present a pathway-based overview. In this review, we suggest a simplified algorithm on how to manage pancreatic neuroendocrine tumors from a practical perspective based on pathologist ’analysis.

**Abstract:**

Current knowledge on the molecular landscape of pancreatic neuroendocrine tumors (PanNETs) has advanced significantly. Still, the cellular origin of PanNETs is uncertain and the associated mechanisms remain largely unknown. DAXX/ATRX and MEN1 are the three most frequently altered genes that drive PanNETs. They are recognized as a link between genetics and epigenetics. Moreover, the acknowledged impact on DNA methylation by somatic mutations in MEN1 is a valid hallmark of epigenetic mechanism. DAXX/ATRX and MEN1 can be studied at the immunohistochemical level as a reliable surrogate for sequencing. DAXX/ATRX mutations promote alternative lengthening of telomeres (ALT) activation, determined by specific fluorescence in situ hybridization (FISH) analysis. ALT phenotype is considered a significant predictor of worse prognosis and a marker of pancreatic origin. Additionally, ARX/PDX1 expression is linked to important epigenomic alterations and can be used as lineage associated immunohistochemical marker. Herein, ARX/PDX1 association with DAXX/ATRX/MEN1 and ALT can be studied through pathological assessment, as these biomarkers may provide important clues to the mechanism underlying disease pathogenesis. In this review, we present an overview of a new approach to tumor stratification based on genetic and epigenetic characteristics as well as cellular origin, with prognostic consequences.

## 1. Introduction

Pancreatic neuroendocrine neoplasms (PanNENs) make up 1–3% of all cases of pancreatic cancer and are the second most common malignancy of the pancreas, after pancreatic ductal adenocarcinoma [1]. PanNENs have a unique clinical presentation as they are highly heterogeneous. Despite all being classified together under a single nomenclature, these composite entities show more differences than similarities although mostly unexplored, as, for example, their cellular origin.

The incidence of these neoplasms has been rising all over the world, from 0.17/100,000 in 1973–1977 to 0.48/100,000 in SEER 18 (Data from the National Cancer Institute Surveillance, Epidemiology, and End Results) (2000–2012), probably in the context of increased incidental detection using endoscopy and cross-sectional imaging [2]. Moreover, pancreatic neuroendocrine tumors (PanNETs) along with the small intestinal NETs tend to be most prevalent NETs in Europe [3], due to the indolent course of some PanNETs.

PanNETs are generally divided into functional (F-PanNETs) and non-functional tumors (NF-PanNETs). In the category F-PanNETs are included the following most frequently tumors: insulinoma, gastrinoma, VIPoma, glucagonoma, and somatostatinoma [4]. However, between 60 and 85% of these neoplasms, are represented by NF-PanNETs. This major category does not secrete hormones but produce peptides which do not generate clinically noticeable effects, being frequently larger in size, hypervascular and often discovered in the metastatic stage [5].

The morphological differentiation splits these PanNENs in three major categories: well-differentiated tumors (PanNETs), poorly-differentiated tumors or neuroendocrine carcinomas (PanNECs), and mixed neuroendocrine non-neuroendocrine neoplasms (MiNENs) [6]. Based on their mitotic count/or Ki-67 labeling index, PanNETs are divided into well-differentiated tumors of low (G1), intermediate (G2), and high grade (G3), while PanNECs are considered G3 by definition [7]. Using the classification of PanNENs as a framework, it has been encouraged to use the next WHO classifications of NENs and ensure a common taxonomy regardless of their site [8]. In addition, the latest WHO classification of NENs outlines the importance of the unique molecular pattern that differentiates PanNETs G3 from PanNECs that will be discussed further in this review [9].

The process of carcinogenesis is thought to be influenced by epigenetic dysregulation due to the amount of dark molecular matter (samples without known driver gene mutations) in PanNETs. Studies using large scale omics techniques revealed further insights into the epigenetic mechanisms correlated with the majority of frequently mutated genes in PanNETs. Explicitly, around 70% of patients present DAXX (death-domain associated protein) or ATRX (α-thalassemia mental retardation syndrome X-linked protein) and MEN1 (multiple endocrine neoplasia type 1) mutations, which encode for proteins involved in the restoration of the aberrations of chromatin-remodeling machinery [10,11]. The high frequency of somatic MEN1 mutations linked to DNA methylation, suggests the substantial role of epigenetics in the origin and development of PanNETs [12]. Therefore, whether Pan-NETs should be regarded as genetically driven malignancies or epigenetically induced neoplasms remains an open topic. The impact of DAXX/ATRX and MEN 1 mutations on tumor biology, methylomes and their association with different phenotypic features, i.e., alternative lengthening of telomeres (ALT) are further detailed in this review. ATRX/DAXX and MEN1 mutations result in loss of nuclear expression of their proteins that can be evaluated by immunohistochemistry (IHC) as a surrogate for mutation analysis [13], although there is no universal agreement regarding the immunohistochemical techniques especially for menin labeling [14]. Furthermore, DAXX and ATRX negative tumors correlate frequently with ALT, a telomerase-independent telomere maintenance mechanism to overcome telomere shortening [15]. This process results in noticeable nuclear foci with high telomere DNA content and significant telomere length heterogeneity within the malignant cells. These characteristics can be clearly identified by telomere-specific fluorescence in situ hybridization (FISH) analysis in archived specimens [15]. Furthermore, recent studies have shown that several transcription factors (TFs) influence the neuroendocrine lineage allocation by progressive accumulation of complex layers of epigenetic modifications [16]. The role of the following TFs, aristaless-related homeobox gene (ARX) and pancreatic and duodenal homeobox 1 (PDX1) in alpha and beta cell differentiation will also be discussed. Due to the fact that their expression can be studied using IHC, these two TFs may be easily implemented in clinical practice. Understanding the influence of epigenetics on the genetic variants [17], as well as on the cancer cell-of-origin, can determine evolution of the disease and the required treatment [18,19]. In consequence, this review presents epigenetic signatures accompanied by genetic alterations that can be applied on a practical level in order to provide a simplified algorithm to optimize PanNETs workup.

## 2. Aim and Strategy

This review explores a recent update on the current knowledge on the genomic and epigenomic alterations and their link with the cells-of-origin of PanNETs in terms of their usefulness as diagnostic or prognostic biomarkers. Further, our review provides a routine algorithm for better diagnosis and cancer risk stratification (indolent versus aggressive), that needs to be validated in further studies.

Published data in the PUBMED and SCOPUS databases served for selective literature research. The following relevant key words as queries, either alone or in combination, were used: ‘pancreatic neuroendocrine neoplasms’, ‘pancreatic neuroendocrine tumors’, ‘genetic landscape’, ‘epigenetic landscape’, ‘origin of pancreatic neuroendocrine tumors’, and ‘classification of pancreatic neuroendocrine tumors’. Additional studies were identified by reviewing the references of all selected articles, while publications from major scientific meetings were searched manually. In total, 194 articles indexed in the English language were scanned, following which, 64 titles were removed because they did not satisfy the subject of the present review. The inclusion criteria were reviews and original articles with different degrees of statistical power due to the rarity of these neoplasms. Exclusion criteria were case reports and publications with abstracts not relevant (case studies removed, 5; publications removed, 59). A total of 130 publications considered potentially eligible were retrieved in full and evaluated. The last update of the research was made in July 2022. Publications from the past 5 years which addressed topics of genomic and epigenomic alterations were prioritized. At the end of the selection process 88 publications were included.

## 3. Origin of PanNETs

Described for the first time in 1869, neuroendocrine cells of the pancreas are embedded in small clusters so-called islets of Langerhans. At this level, five different well-defined cell types are classified as A cells or α (secrete glucagon), B cells or β (produce insulin), D cells or δ (produce somatostatin), PP cells (secrete pancreatic polypeptide) and epsilon cells (secrete ghrelin). Although the secretion of different hormones is suggestive for the cell origin, it is an unanswered question whether it also indicates the origin of NF-PanNETs [20].

PanNETs were previously known as islet cell tumors as they were believed to originate from these structures. Later studies reported that they may derive from the ductal system where neuroendocrine cells aggregate and proliferate in the context of nesidioblastosis phenomenon [21]. These findings suggest that neuroendocrine cells of these tumors arise from both sources. Moreover, referring to the possible origin in the ductal system [22,23], PanNETs show significant genetic differences when compared to pancreatic adenocarcinomas [24]. There is also evidence that in patients who present primary tumor alterations like MEN1 mutations, also frequently display an accumulation of these mutations in hyperplastic endocrine cells that become dysplastic and probably capable to progress into neoplasms [23,25]. Recently, an alternative paradigm has been proposed, that these tumors are more likely to develop by dedifferentiation from mature cells with neuroendocrine phenotype, which accumulate mutations and are able to proliferate [25].

The development of pancreatic cells is coordinated in a spatial and temporal way by a series of cell-type-specific TFs [26], namely: PDX1, ARX, NeuroD1/BETA2 (proneural basic helix-loop-helix), NEUROG3 (neurogenin 3), ISL1 (insulin gene enhancer 1), INSM1 (insulinoma-associated 1), PAX6 (paired box gene 6) and NKX 2.2 (homeobox protein). The expression of certain TFs is also maintained at the level of PanNETs, rendering possible the detection of the corresponding origin of the tumor. Their expression is induced by specific epigenetic markers situated at the relative regulatory sites (e.g., super-enhancer activation) [27]. Super-enhancers are a region of mammalian genome comprising large clusters of enhancers which are linked to TFs. They play a crucial role in the regulatory mechanisms by which different cells form specific gene expression programs [27]. Super-enhancers describe cell-specific chromatin signatures defined by H3K27 and H3K4me1/2, where cell identity is encoded. Thus, lineage-specific super-enhancers may be involved in sustaining tumor cell origin [16]. Furthermore, transdifferentiation between α and β cells in embryos influences endocrine lineages, while lineage decision results from cross-inhibitory implications [28]. NEUROG3 generates the endocrine lineage, with mutual inhibition between opposing lineage determinants necessary for the differentiation of α and β cells, respectively [29]. NKX6.1, PAX4, and PDX1 TFs influence β-cells fate over α cells. On the other hand, ARX gain of function leads to an increase in α cells and a decrease in β cells [30,31,32]. ARX and PDX1 TFs are expressed in different locations during islet cell development. Specifically, the former is expressed in endocrine precursors, with different endocrine cell types sharing ARX expression [33], while PDX1 expression is found in differentiated β-cells [15]. Additionally, TFs in PanNETs are expressed differently depending on the dominant hormone production. Consequently, specific NETs are produced by hormonally driven cells affected by specific mutational events [34]. Further evidence in favor of this theory comes from the expression of PDX1 that is particular to cells in the phase of initiating insulin expression and remains up-regulated solely in β-cells of the pancreatic islets of Langerhans [35].

It is important to notice that the implication of the epigenetics in ontogeny drives the susceptibility for particular genetic driver mutations, revealing a different group of genes connected to lineage and tumorigenesis. Therefore, finding the cell-of-origin is decisive to recognize the tumor biology and progression [19]. The status of cell type ARX and PDX1 TFs offers an insight into epigenetic processes that interfere with neuroendocrine lineage designation [16,36] In the literature, the presence of ARX and PDX1 in 84% of NF-PanNETs [16], was described and might be categorized as “A-type” (similar to α cells) or “B-type” (similar to β cells) [37]. PanNETs of subtype A and B have super-enhanced ARX and PDX1 regions as they influence the α and β cell signature. Furthermore, IHC for ARX and PDX1 expression can be used for subtyping, as subtypes showed significant difference in relapse free survival (RFS) [16]. Specifically, while the expression of PDX1 is particular for low grade and benign tumors, mainly insulinomas, ARX expression is specific to advanced stages. In consequence, their status might be used as a surrogate marker for these cellular subtypes for an improved risk stratification. Clinically, PDX1 expression is more practical than ARX+, with patients with small PDX1+ tumors only needing conservative following, while patients with PDX1− tumors may be prone to early metastases amenable to surgical or medical treatment and require vigilant monitoring. ARX expression is correlated with ALT+ status and offers mainly prognostic information [38]. 

The DNA methylation analysis (this term will be discussed in more detail on Section 5. Molecular landscape) supports the identification of tumors subtypes as a result of cancer cell’s epigenetic memory regarding their origin [39,40]. For the first time, Gita Boons et al. [37] demonstrated that compared to regular pancreatic islets, Pan-NENs exhibit a distinct genome-wide DNA methylation profile. Thus, in addition to PDX1 expression, the study of DNA methylation of the PDX1 gene region might be utilized to differentiate between the two PanNENs A and B subgroups and indicate a connection to each subtype’s individual cell of origin, α or β cells. The two subgroups appear to have a distinct molecular profile.; specifically, subgroup A as NF-PanNETs with noticeably worse outcome, while subgroup B, represented by insulinomas, have a good prognosis and are usually not methylated at the PDX1 promoter. However, in rare circumstances, α cell-like NF-PanNETs may transdifferentiate by developing the capacity to secrete insulin. This subgroup of insulinomas presents ARX+ cells, ALT activation and has a metastatic behavior. Their molecular profile is often seen on NF-PanNETs, as ATRX/DAXX mutations were described. In contrast, typical indolent insulinomas present a different genetic background than NF-PanNETs. Therefore, the genetic and epigenetic features can differentiate between typical indolent insulinomas and aggressive insulinomas [41,42]. In addition, in a recent study, it was shown that ATRX/DAXX/MEN1 (A-D-M) mutated tumors and A-D-M wild type (WT) cluster differently based on DNA methylation and gene expression patterns. As DNA promoter methylation inhibits transcription by changing the accessibility of binding sites for transcription activators, reported DNA methylation levels were high in the PDX1 promotor in A-D-M mutated tumors. Thus, mutant tumors display a high ARX and low PDX1 expression compared with WT tumors and the methylation profile is similar for A-D-M-aberrant PanNETs and α cells [36]. An additional study reported the *PDX1* promotor was hypermethylated in α cells while that the DNA methylation levels were low in the PDX1 promotor in β cells [43].

Summarizing, the A-D-M mutant PanNETs subtype reported by Chan et al. [36] and the type A subtype reported by Cejas et al. [16] are corroborated by the discovery that A subtype of PanNETs high in ARX showed similarities to pancreatic α cells and enrichment of mutations in A-D-M. In contrast, the B subtype of PanNETs that is high in PDX1, resembles to pancreatic β cells with a A-D-M WT tumor profile, but with more variable levels of PDX1 expression [44]. 

## 4. Pathology and Molecular Subtypes

The classification system of NENs has evolved considerably for the last 20 years. Following the WHO 2000 classification, the well-differentiated endocrine tumor nomenclature was specifically introduced to replace many distinct names based on potential cells of origin. A further relevant step of international standardization in the gastro-entero-pancreatic-NENs system was achieved with the WHO 2010 classification, where their diagnosis was separated from staging and grading. NENs were introduced as the combination of two genetically and biologically distinct entities, i.e., well-differentiated NETs and poorly-differentiated NECs. The difference between these two histological tumor classes (NET versus NEC) has clinical as well as evolutionary importance. Those that are well-differentiated have indolent character and better evolution, compared to the poorly-differentiated ones, which can be more aggressive with worse prognostic [45,46]. Further, well-differentiated NETs with high proliferation rate (NETs G3) were first added to the revised 2017 classification of PanNENs [7]. This recent subgroup of PanNENs G3, NET G3, presents driver mutations comparable to those found in NETs, and shows less responsiveness to platinum-based therapy and longer median survival [47]. Presently, in order to make a differential diagnosis between NET G3 and NEC, it is necessary to confirm their neuroendocrine differentiation in the context of the proper morphology, besides IHC [9].

Another important tool for the classification of NENs is the grade of proliferation of neoplastic cells evaluated by the mitotic count and/or Ki67 labeling index. The grading of these neoplasms is considered an important prognostic factor, independent of tumor stage and describes how a biologically aggressive these neoplasms are [48]. The mitotic rate and KI-67 proliferation index are determined using IHC measured in the most mitotically active areas of the pathological specimen. In Pan-NENs, the accuracy of the measurement of both mitotic rate and KI-67 index in a single biopsy or tumor specimen is susceptible to sampling or interpretation errors due to the intratumoral heterogeneity and the lack of method standardization [49]. Moreover, the IHC evaluation is labor-intensive and semi-quantitative, being based on the subjective evaluation of the examiner, and, thus, can affect the accuracy of grading and classification. Before surgical intervention, most Pan-NENs are often diagnosed using ultrasound-guided fine-needle aspiration biopsy (FNA-B). In FNA-B samples, the quantification of KI-67 proliferation index is represented by the need of at least 500 cells in the regions of highest labelling (hot-spots). Tumor hot-spots are identified as regions with a higher level of nuclear Ki-67 staining. It has been demonstrated [50] that the Ki-67 count decreases as the hotspot size increases, underscoring the significance of standardizing this measure for accurate assessment. In addition, it is challenging to pinpoint not just the hotspot’s size but also its shape. Thus, as FNA-B samples often present pauci-cellularity, this may constitute a limitation for the correct detection of KI-67 and tumoral grading, especially in small (<2 cm) or advanced Pan-NETs, where a non-surgical approach is applied and/or where a neoadjuvant therapy is planned [51]. Thus, current markers for tumoral grading in Pan-NENs samples showing intratumorally heterogeneity and/or pauci-cellularity present many limitations in terms of accuracy of diagnosis and in defining the tumoral behavior. Therefore, an upgraded classification into tumor subtypes would emerge from an improved definition of PanNENs given by a molecular signature.

For example, since the differential diagnosis between a NET6 G3 and a NEC in the pancreas may not always be clear, IHC can help with biomarker screening based on the mutational profile of PanNETs and PanNECs. Mutations in DAXX or ATRX genes are good flagships as roughly 40 to 50% of the PanNETs contain them and they can be detected by IHC [52], while features of PanNECs include aberrant p53 expression (TP53 inactivation) and the loss of Rb1 [53,54,55], but never mutations in DAXX or ATRX [46,56] (Table 1). p53 and Rb proteins can be used as markers of NECs in all sites, while the lack of DAXX and ATRX mutations in NENs subsets other than pancreatic NENs limits the use of the complete approach to this type of tumor only [57].

A recent study performed by Simon et al. [58] outlined the importance of finding molecular groups by performing genetic and epigenetic profiling of 57 PanNENs. The molecular classification accurately differentiates PanNECs that may have an exocrine cell-of-origin, from PanNETs’ endocrine cell-of-origin. Explicitly, MEN1, DAXX, ATRX, VHL, PTEN, and TSC2 are frequently linked to PanNETs, while the group of PanNECs includes mutations in KRAS, SMAD4, and TP53, but no mutations in DAXX, ATRX, or MEN1. In addition, a recent multi-omics analysis [44] identified in a non-selected cohort of PanNENs, a new proliferative subgroup consisted of roughly equal distribution of PanNETs and PanNECs. This new subcategory with high cell proliferation suggests that from a molecular point of view, some PanNETs resemble to PanNECs, sharing histopathological similarities with other well-differentiated tumors.

## 5. Molecular Landscape

The majority of PanNETs are sporadic (90%), but can also be associated with other hereditary syndromes (10% of cases) like MEN1 (multiple endocrine neoplasia type 1), VHL (von Hippel Lindau) syndrome, NF1 (neurofibromatosis type 1) [59]. Recent studies using high-throughput next-generation sequencing methods aim to split these tumors in subgroups according to their molecular characteristics for improved risk stratification. Only around 40% of sporadic PanNETs are attributed to mutations alone as the cause of carcinogenesis; the remainder are assigned to epigenetic changes. The frequent inactivation of A-D-M mutations suggests a potential role of the epigenetic mechanisms of PanNETs which is worth studying, especially in the case of sporadic tumors, which are more aggressive compared to those considered part of familiar syndromes (although initially it was thought that the association of MEN 1 and ATRX/DAXX mutations would have a good prognostic [24]). The effects of A-D-M mutations on PanNETs’ development and how DNA methylation, histone changes, and chromatin remodeling control these genes’ expression as part of the epigenetic process will be discussed in detail below.

### 5.1. Genetic Basis: A-D-M Mutations

Potential driver mutations mostly affect tumor suppressor genes and rarely proto-oncogenes, making PanNETs a ‘‘tumor-suppressor’’ disease [11,24]. The level of mutational burden is very low in all well-differentiated NETs. Still, the highest rate of detectable driver mutations was reported in PanNETs and belong mostly to key genes involved in altered telomere length, chromatin alteration, mTOR signaling and DNA damage repair pathways [11]. The somatic mutations associated to sporadic PanNETs most frequently are in MEN1 (44%), DAXX (25%) and ATRX (17.6%) and in the genes involved in mTOR signaling pathway (15%) [24].

In 2011, a landmark paper described the somatic mutations involved with sporadic PanNETs, which occur in two ways; the first one is involved in chromatin remodeling and includes MEN1, a nuclear protein part of a histone methyltransferase (“HMT”) complex, the modifying factor of chromatin ATRX, as well as the regulator factor of apoptosis, DAXX. The second way includes the genes involved in the mTOR signaling pathway, as well as TSC2 (tuberous sclerosis complex), PTEN (phosphatase and tensin homolog), and PIK3CA [24]. Subsequently, another extended study defined four major molecular ways that determine the PanNETs development ± progression, namely: chromatin remodeling (MEN1, SETD2 (Set Domain Containing 2), ARID1A (AT-rich interactive domain-containing protein 1A) and MLL3 (myeloid/lymphoid or mixed-lineage leukemia protein 3)), DNA damage repair [(MUTYH (mutY DNA glycolase), CHECK2 (checkpoint kinase 2) and BRCA2 (Breast Cancer type 2)] activation of the mTOR signaling pathway [(TSC1, TSC2, PTEN and DEPDC5 (DEP Domain Containing 5)], and telomere maintenance (DAXX, ATRX) with a subgroup of PanNETs being associated with hypoxia and the HIF (hypoxia inducible factor) signaling pathway. Additionally, MEN1 influences all these key processes [11]. Similarly, as part of this study it was demonstrated that 17% of apparently sporadic tumors exhibit a significant proportion of germline events, including mutations of the genes involved in DNA damage repair. 

MEN1 is considered the most frequently altered gene in sporadic PanNETs, as somatic mutations of MEN1 have been reported in 25 to 44% of PanNETs [12]. MEN1 serves as a tumor suppressor gene in both sporadic and syndromic PanNETs, with loss of function of both alleles [60]. The MEN1 gene is located on chromosome 11q13 and is implicated in early embryogenesis due to the variety of MEN1-related lesions and the disparate embryonic origins of afflicted organs. Comparative genomic study of tumoral and constitutional genotypes has revealed evidence of somatic loss of heterozygosity (LOH) for the MEN1 gene [61]. This is consistent with the hypothesis that the development of MEN1-associated cancers occurs in two stages, where MEN1′s first allele is affected by the germline mutation, and a second somatic inactivation of the unaffected allele (LOH) [62]. Due to MEN1′s association with the MEN1 syndrome, it is well known that MEN1 codes for menin, a nuclear protein with a crucial role in chromatin remodeling control. It influences growth and differentiation of neuroendocrine cells by regulating histone methylation in promoters of specific target genes [63].

DAXX/ATRX mutations are mutually exclusive, which is consistent with their function as part of the same pathway [24]. DAXX function is essential for several nuclear functions, as for example transcription and cell cycle regulation (including apoptosis) [64]. One of the DAXX copies is inactivated by mutation and the other by loss or epigenetic silencing. One of the ATRX copies is inactivated by mutation and other by chromosome X inactivation. Both proteins are needed for incorporation of histone variant H3.3 at the telomeric ends of chromosomes [65], a process that is supported by nuclear staining of the proteins in DAXX/ATRX WT PanNETs, which is missing in ATRX/DAXX mutated PanNETs [24,66]. Despite almost exclusively being associated with PanNETs within NENs, DAXX/ATRX mutations are correlated with a shorter disease-free survival in comparison to those with WT [64].

### 5.2. Epigenetic Basis: A-D-M Status in DNA Methylation

Epigenetic modifications in PanNETs, which are usually associated with a high rate of CpG hypermethylation, are determined by genome-wide DNA methylation (methylome) profiling investigations. This analysis is going through a transformation akin to that seen with the introduction of high-throughput tools for DNA sequencing. Through the methylation of promoters and enhancers, DNA methylation controls gene expression. As a result of promoter hypermethylation and downregulated gene expression in cancer cells, tumor-suppressor genes are silenced, which promotes the growth of tumors. Genome-wide hypomethylation is another sign of cancer and has been linked to increased chromosomal instability (CIN). Methylating and demethylating enzymes control the dynamic process of DNA methylation. DNA n-methyl transferases (DNMTs) modify the cytosine in CpG DNA sequences by adding a methyl group; demethylating enzymes, which also include members of the ten-eleven translocation (TET) enzyme family, take away the methyl group [65]. 

DAXX and ATRX influence DNA methylation, with DAXX specifically binding to and directing DNMT to the RASSFA promoter [67]. It was reported that PanNETs with DAXX or ATRX mutations had different levels of methylation compared to tumors without mutations, with DAXX mutated patients exhibiting more pronounced changes [68]. This finding is extremely intriguing considering how DAXX and DNMT interact as well as the fact that PanNETs are the only population in which DAXX mutations are more common than ATRX mutations [54,69].

Furthermore, MEN1-related PanNETs likewise have a high rate of promoter and genome-wide hypermethylation that is considered a distinctive feature of MEN1-related neoplasms [12,65]. In addition, MEN1 deficiency was reported to be associated with CpG hypermethylation contrary to sporadic or VHL-associated PanNETs [70]. These findings support the role of DNA methylation in PanNETs as being influenced by MEN1 mutations and may explain why MEN1 syndrome has inconsistent genotype-phenotype relationships [71].

To gain insights into this tumor type’s epigenetic underpinnings, three subgroups of PanNETs (termed T1, T2 and T3) were devised using DNA methylation analysis with corresponding clinical and genomic data [71]. The first subgroup (T1) exhibited F-PanNETs and A-D-M WT genotypes, which is consistent with the findings of Boons et al. [37] and Chan et al. [36]. Specifically, gene expression and methylation profiles are less homogenous in WT tumors than in mutant A-D-M tumors. Tumors with mutations in A-D-M included in the T2 subgroup, as well as recurrent LOH across chromosomes, have been linked to a lower survival rate in PanNETs [11,64], suggesting that both LOH and methylation may play a role in carcinogenesis in PanNETs. The T3 subgroup included mutations in MEN1, was enriched for G1 tumors and exhibited histological features linked with a better prognosis. Additionally, ARX and PDX1 were proposed to distinguish A-D-M mutants from WT PanNETs [36,71]. ARX methylation patterns was investigated individually because the ARX gene is located on the X chromosome, which is generally excluded from methylation analyses, but there were no variations in methylation levels across the groups. However, T2 and T3 subgroups had higher gene expression than T1 (WT tumor). This is consistent with earlier research [16,36] and implies that the methylation status of ARX does not impact the gene’s expression. In T2 and T3 subgroups with A-D-M mutant tumors, PDX1 presented lower expression and hypermethylation than T1 subgroup with WT tumors. As PDX1 was suggested to mark pancreatic β-cells [72], T1 tumors might have originated from these. Overall, these novel findings [71] contribute to an improved understanding of PanNETs’ genetic and epigenetic landscape, showing that methylation patterns may be used to stratify PanNETs prognosis, however the link between methylation event of genes and gene expression requires further study.

Another study described three other tumor groups that can be separated upon genetic and epigenetic evolution: α-like, β-like and intermediate (-ADM and -WT) PanNETs. These subgroups form as follows: the development of α like tumors from α-cells is enhanced by MEN1 mutation; tumor progression is associated with intermediate tumors gradually losing differentiation due to DAXX/ATRX mutations, chromosomal differentiation instability with recurrent LOH and ALT activation; β-like tumors develop from β-cells and usually manifest as insulinomas with indolent course and distinct genetic background [19]. Early-stage tumors, with MEN1 mutation, exhibit clear α-like epigenetic features and favorable outcome, while relapses and metastases occurred mostly in both intermediate PanNET subgroups [19]. Identifying intermediate groups of PanNETs could be improved by using methylation analysis of ARX and PDX1. In the case of intermediate-WT tumors, their expression was found to be equally distributed while DNA-methylation profile indicates similarity with β-cells [19].

### 5.3. Epigenetics Basis: A-D-M and ALT Status in Chromatin Remodeling, Histone Changes and Telomere Alteration

Histone proteins bind around DNA to form nucleosomes, which subsequently compact to generate chromatin. Histone proteins can be altered by adding acetyl, methyl, phosphoryl, or ubiquitin groups to particular residues, frequently found in the N-terminal histone tails. Therefore, TFs no longer have the same affinity binding chromatin, which affects how genes are expressed. Normally, the histone complex has one copy of H1, two copies of H2A, H2B, H3, and H4. A family of linker histones termed histone H1 is involved in chromatin maintenance, controlling gene expression, and chromatin-based DNA repair [73].

MEN 1 is involved in chromatin remodeling as a part of a multiple protein complex with a histone H3 lysine 4 methyltransferase activity and interacts with various histone deacethylases and histone methyltransferases (e.g., PRMT5 and SUV39H1) to either repress or activate gene transcription. Covalent histone changes could play a role as possible propagators of disease pathogenesis in PanNETs due to the recurrent mutations of MEN1 in sporadic PanNETs [63]. Additionally, the role of mutations in DAXX and ATRX proteins as promoters of tumorigenesis can be understood through their involvement in chromatin remodeling, similar to menin, especially in the telomeric areas [65].

Further confirmation of DAXX/ATRX mutated tumors being a more aggressive subtype comes from the fact that the ALT mechanism is frequently associated in 6–21% of PanNETs with these mutations [11,15,54,64,74]. ALT is also present in immortalized cell lines and in a growing number of tumors, such as pituitary adenoma, neuroblastoma, glioma, and uterine leiomyomas [75,76,77,78]. The association between ALT and mutations in DAXX/ATRX in NF-PanNETs was initially noted by Heaphy and colleagues, where 61% of PanNETs were found to display abnormal telomeres specific to ALT [15]. Furthermore, by using FISH technique, 19 tumor samples with ATRX/DAXX mutations were all found to have abnormal ALT levels. Moreover, ATRX/DAXX mutation and the presence of ALT were confirmed in 33/98 PanNETs by Scarpa et al. [11]. Strong correlations between the ALT status and abnormal nuclear ATRX/DAXX expression were associated with lymphovascular invasion, perineural invasion, lymph node involvement, distant metastases, and shorter recurrence-free survival [64]. According to later extended retrospective studies, ATRX/DAXX mutations and the ALT phenotype appear as are late events of PanNETs within MEN-1 syndrome [79] and they are associated with CIN in PanNETs [52]. Additionally, ALT status can be applied as a high specific diagnostic biomarker in cases of a NET metastasis of unknown origin to verify the possibility of a pancreatic primary [80,81].

Based on the findings of Hackeng et al. [82], ATRX/DAXX loss and ALT positivity are independent, unfavorable prognostic biomarkers for RFS for small NF-PanNETs (≤2.0 cm) and without metastases at regional lymph nodes. Moreover, the implementation of ALT status and DAXX/ATRX expression as a routine tool in the pre-operative decision might be facilitated by the opportunity to accurately determine ALT status using FISH, and DAXX and ATRX expression using IHC as prognostic biomarkers of NF-PanNETs [80,81,83].

### 5.4. Pathway-Based Interaction of PanNETs

As aforementioned, mTOR/AKT/PI3K pathway is a well-established core pathway involved in the pathogenesis of PanNETs. mTOR/AKT/PI3K pathway plays a central role in tumor oncogenesis and development. Mutations in the mTOR/AKT/PI3K pathway appear in 14.7% of cases with sporadic PanNETs. In particular, mutually exclusive mutations in PTEN (7.1%), TSC1 (2.0%), TSC2 (2.0%), and in DEPDC5 (2.0%) were discovered [11]. The majority of these genes suppress the mTORC1 complex or its subsequent activation of HIF, and their mutation was correlated with poorer prognosis. In view of these observations, transcriptional findings support the relevance of the mTOR pathway, with TSC and PTEN negative-regulation being key events in a significant proportion of PanNETs. PTEN is involved in the inhibition of mTOR/AKT/ PI3K pathway, therefore PTEN mutation, with subsequently increased activity of this pathway, was linked with more aggressive PanNETs and poorer survival [54].

Interestingly, the effects of ATRX/DAXX mutation extend further than the ALT mechanism and may involve the direct control of PTEN gene. This pathway-based interaction is observed from a mechanistic approach in PanNETs tumorogenesis [54]. Specifically, MEN 1 orchestrates directly or indirectly all core pathways acting as a “hub” gene (Figure 1). Thus, menin plays a key role in the crosstalk between mTOR/AKT/PI3K pathway and the chromatin remodeling compartment. As MEN1 and PTEN co-mutation has been described in a limited number of PanNETs [11,24], it was found that menin and PTEN inhibition of the mTOR/AKT/PI3K pathway may cooperate to prevent tumor proliferation. Furthermore, menin may function in tandem with DAXX to repress matrix metalloendopeptidase (MME), a zinc-dependent metalloprotease necessary for PanNET cell growth, suppressing PanNETs tumorigenesis. DAXX also coordinates PTEN location between the nucleus and the cytoplasm, whereas PTEN is involved in the regulation of DAXX’s gene expression [84,85]. As these genes are strongly connected to epigenetic changes (DNA methylation, histone modification and chromatin remodeling), their alteration promotes cancer growth through the mTOR pathway. Therefore, the integrated epigenetic and genetic data resulted from this range of interactions between altered components of different pathways may be interesting for future studies.

## 6. Discussion

Recent studies have shown that molecular landscape found in PanNENs integrates the mutational and epigenetic profiling. These findings distinguish different subgroups based on epigenetic signatures of endocrine differentiation. Thus, the epigenetic background mirrors specific cells-of-origin of PanNETs. The epigenetic subtypes resemble to mature alpha, beta, or intermediate cells and are specialized for structural changes, mutational patterns, and prognostic significance. However, additional research using high-throughput technologies is required to completely unify data on aberrant pathways.

In summary, based on the presented data [9,19,41,42,44,71,82], at least five groups of PanNENs can be identified as they differ by cell of origin, clinical features, genetic, and epigenetic background (Table 2).

The first group, PanNEN 1, refers to early-stage tumors, with MEN1, but no ATRX/DAXX mutations and favorable outcome. These also exhibit clear α-like epigenetic features and high expression level of ARX. The second group, PanNEN 2, includes tumors mutated in A-D-M with an intermediate epigenetic profile but preponderantly α-like features. Their high susceptibility for ALT activation sustains their poor prognosis. The following two groups (PanNEN 3 and 4) uncover the differences between typical insulinomas with an indolent course and aggressive insulinomas. Typical insulinomas resemble to β-cells with high expression of PDX1 and a good prognosis. In addition, these indolent neoplasms differ from the other subgroups in terms of their genetic background [9]. In contrast, PanNEN 4 resembles more likely to α cells, one theory suggesting that they probably existed as a NF-PanNET before developing functional features [42]. The last subtype refers to a proliferative group proposed by Kevin Yang et al. [44]. This new subgroup has both well-differentiated and poorly differentiated PanNENs which is a sign of increased cell proliferation and inferior survival. While the PanNENs molecular profiling distinguishes different tumor subtypes, some of the major challenges is to translate this finding in clinical practice, as well as recognizing a subgroup more likely to be responsive to a specific targeted therapy.

At present, many NETs are treated similarly while NECs are treated mainly in parallel to data from lung tumors. Primary tumor surgery is associated with the reduction of metastasis risk, and the tumor size is the only pre-surgery marker that contributes to the therapeutic decision. The other prognostic markers for metastasis (microscopic invasion, number of lymph nodes affected, perineural invasion and tumor stage) are determined post-surgery [86]. Further, surgery is the cornerstone treatment for local or locoregional disease, although for patients with R0 resection the 5-year disease free survival is approximately 50% [87]. The advanced tumors are usually treated with surgery complemented by drug therapy. The implementation of adjuvant therapy for patients at high risk of recurrence is limited now by the lack of an approved risk prediction tool to guide follow-up and patient selection. The strongest indicator of recurrent disease after surgery has been proven to be ALT + and/or ATRX-/DAXX- [82]. Thereby, therapy indication and response could benefit from the feasible and existing molecular classifications in addition to the established morphological classification.

Integrating the existing knowledge presented in the review, we propose the implementation of a routine algorithm in PanNETs’ management. This algorithm (Table 3) may improve risk stratification, as the status of all six biomarkers may indicate different cellular types of PanNETs with different outcomes; therefore, the therapeutic interventions will differ among these subgroups toward a better survival. The proposed algorithm presents a broad feasibility to accurately determine the following biomarkers. As aforementioned, immunolabelling for DAXX/ATRX/MEN1 and for ARX/PDX1 using materials from FNA-B or formalin-fixed paraffin-embedded tissue sections is a reliable method for studying their mutational status. Moreover, IHC remains the most convenient method for diagnosing NENs and proliferation factors. The advantages of this technique arise from the affordable cost and the study of archived paraffin blocks or biopsies. Furthermore, besides FISH analysis, there are currently several other methods to detect ALT status as for example terminal restriction fragment (TRF) Southern-blot analysis of telomere length profile, combined promyelocytic leukemia (PML) immunofluorescence/telomere FISH analysis of tumor sections for ALT associated PML or using C-circle amplification assay [88].

On a practical level, ALT can play a double role, first by reliably indicating patients with “high-risk” PanNETs and secondly as a very specific marker of pancreatic origin for neuroendocrine metastasis from unknown primary. An interesting issue for future perspectives is the relationship between ATRX/DAXX mutations with ALT activation and α cell like characteristics of NF-PanNETs. Although there are no clinically validated treatment approaches to target ATRX/DAXX mutations in PanNETs, this is an active field of research because of the associated ALT phenotype’s molecular peculiarity.

Another point of interest is the analysis of the cells of origin of these tumor subtypes, which could clarify their unpredictable evolution and potentially facilitate a new target therapy. The majority of studies presented in this review included preponderantly sporadic tumors, with few MEN1-syndrome related NF-PanNETs, although it is not certain if their findings apply to MEN-1 related and other hereditary NF-PanNETs [42].

However, there are still several knowledge gaps that need to be addressed, regarding the translational potential of molecular signatures to the clinical management of PanNETs. For example, using non-invasive methods like liquid biopsy for PanNETs, or the use of cell-free DNA or circulating tumor DNA may predict the clinical behavior of these heterogenous neoplasms. Future directed studies aim to evaluate the cell-of-origin epigenetic markers in order to selectively treat patients who are at increased risk for metastatic NF-PanNET progression and, contrarily, to monitor patients with reduced risk.

## 7. Conclusions

This review presents recent knowledge regarding the genetic and epigenetic alterations associated with PanNETs: driver gene mutations involved in PanNETs with focus on ATRX/DAXX/MEN1 mutations, and the poor prognostic implications of ALT and its association with ATRX/DAXX mutations, as well as the α and β cell signature in PanNETs, recognized by ARX/PDX1 enhancer status. These findings serve as a pivot for epigenetic dysregulation as a key player in PanNETs origin, evolvement, and treatment response. Further, it is anticipated that a multi-omics strategy supported by clinical trials will impact the planning of individualized therapy, prognostic stratification, and diagnostics.

## Figures and Tables

**Figure 1 cancers-14-04444-f001:**
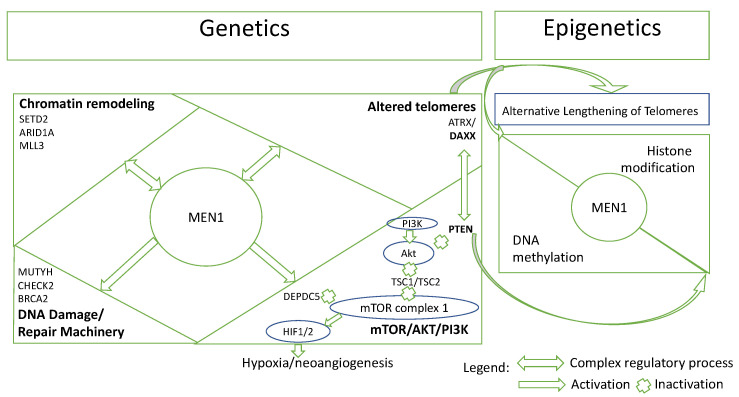
Simplified overview of altered components of main pathways in pancreatic neuroendocrine tumors and their interaction with epigenetic changes mentioned in this review ^1^. ^1^ Please note that this is a simplified figure based on [1,11,54].

**Table 1 cancers-14-04444-t001:** Generalized differential diagnosis between PanNETs and PanNECs based on IHC analysis ^1^.

	P53	Rb1	DAXX	ATRX
PanNETs	−	+	−	−
PanNECs	+	−	+	+

^1^ Please note that this is a simplified algorithm based on [56] that does not absolutely differentiate all PanNETs from PanNECs, as, for example: aberrant p53 expression in PanNEC includes diffuse positivity or total loss of p53; PanNETs—pancreatic neuroendocrine tumor, PanNECs—pancreatic neuroendocrine carcinomas, IHC immunohistochemistry.

**Table 2 cancers-14-04444-t002:** Schematic indicators for pancreatic neuroendocrine neoplasms classification using genetic and epigenetic signatures ^1^.

	PanNEN 1	PanNEN 2	PanNEN 3	PanNEN 4	PanNEN 5
Cell of origin epigenetic markers	α- cell-like (ARX+/PDX−)	α- cell-like > β- cell-like	β- cell-like (ARX−/PDX+)	α- cell-like > β- cell-like	Unknown
Clinical features	Non-functional	Non-functional	Functional-insulinoma	Functional-insulinoma	Non-functional
Genetic signature	MEN1	MEN1/ATRX/DAXX	Other	ATRX/DAXX	Other
Epigenetic pattern	Well-differentiated	Intermediate differentiation	Well-differentiated	Intermediate differentiation	Well and poor differentiated
ALT activation	No	High susceptibility	No	High susceptibility	Unknown
Outcome	↑	↓	↑	↓	↓

^1^ Please note that this table presents schematic indicators based on [9,19,41,42,44,71,82] PanNEN—pancreatic neuroendocrine neoplasm. ↑ favorable; ↓ poor.

**Table 3 cancers-14-04444-t003:** Routine tool in Pancreatic neuroendocrine tumors for risk stratification based on IHC and FISH analysis ^1^.

Risk Stratification	Type of Cell	A-D-M	ARX	PDX1	ALT
Aggressive ++ NF tumor	α like	A-D-M−	+/DN	−	+
Aggressive NF tumor	α like > β like	M−	+/DN	−	−
Aggressive-F tumor (insulinoma)	α like > β like	A-D−	+/DN	−	+
Indolent-F tumor (insulinoma)	β like	A-D-M+	−	+/DP	−

^1^ Please note that this is a simplified algorithm for risk stratification based on references [9,19,41,71,82]; this algorithm does not absolutely identify all the subtypes described since there is no universal agreement especially regarding menin labeling. A-D-M- ATRX/DAXX/MEN1 mutations, DN-double negative (ARX−PDX1−), DP-double positive (ARX+PDX1+); F-functional, FISH-fluorescence in situ hybridization, IHC-immunohistochemistry, NF-nonfunctional.

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
