# Peer review of "Insights into Epigenetic Changes Related to Genetic Variants and Cells-of-Origin of Pancreatic Neuroendocrine Tumors: An Algorithm for Practical Workup"

_cancers, 2022, doi:10.3390/cancers14184444_

Round 1

Reviewer 1 Report

The present study aimed at reviewing recent data on genetic and epigenitic features of neuroendocrine tumors. The topic is of interest and owns a certain novelty. The methods should be better detailed. The introduction and discussion sections seem disproportionally too long and too short, respectively. In particular, the discussion should be better focused and targeted on the clinical implications of the findings of the study (parts of these implications such as costs and clinical availability were already reported in the introduction). Overall, the result section lacks pragmatism and should be better rationalized. Certain topics are recalled in different/separate paragraphs, even sometimes in a non-logical sequence (details are provided before baseline information). This does not allow the reader to progressively build up a clear and net understanding of each aspect analyzed through the paper.  Diagrams or schematic figures would be very helpful. The English writing style should be improved, shortening the length and simplifying the structure of the sentences

Reviewer 2 Report

The review of Oana Ciobanu and colleagues comprises a full text analysis and summary of 87 publications from PubMed and SCOPUS databases. Mutations of the genes ATRX/DAXX/MEN1, which are associated with the mTOR signaling pathway and associated with DNA methylation, are particularly in focus of the study. The mTOR/AKT/PI3K signaling pathway as a whole should be explained in more detail in a short section and here, if possible, the individual components (including PTEN) should also be shown graphically and the significance in terms of epigenetic changes put into context.  

The simple summary is difficult to understand for people who are not familiar with the topic. Here, if possible, do not use abbreviations that are mentioned within the summary and keep it as simple as possible to introduce inexperienced readers.

Overall, it is a nice overview as far as I can tell. Some terminology, such as "superenhanced" (line 200) should be explained. What is a superenhancer and what is the consequence.

What is missing is a visualization of the presented correlations / the algorithm. Table 2 already gives an impression, but could be further elaborated (also in light of the cited paper "Proteotranscriptomic classification and characterization of pancreatic neuroendocrine neoplasms" by Kevin Yang and colleagues). A schematic overview of subtypes based on "cell of origin", "clinical features", "genetic background" and "epigenetic background" would also be beneficial.

Round 2

Reviewer 1 Report

The Authors have undertaken a meticulous and accurate revision, approproately addressing and fixing the issues outlines